# Decrease of Pneumococcal Community-Acquired Pneumonia Hospitalization and Associated Complications in Children after the Implementation of the 13-Valent Pneumococcal Conjugate Vaccine (PCV13) in Taiwan

**DOI:** 10.3390/vaccines9091043

**Published:** 2021-09-18

**Authors:** Ching-Fen Shen, Ju-Ling Chen, Chien-Chou Su, Wen-Liang Lin, Min-Ling Hsieh, Ching-Chun Liu, Ching-Lan Cheng

**Affiliations:** 1Department of Pediatrics, National Cheng Kung University Hospital, College of Medicine, National Cheng Kung University, Tainan 70101, Taiwan; drshen@mail2000.com.tw (C.-F.S.); mactep_8@hotmail.com (M.-L.H.); liucc@mail.ncku.edu.tw (C.-C.L.); 2Institute of Clinical Medicine, College of Medicine, National Cheng Kung University, Tainan 70101, Taiwan; 3Department of Pharmacy, National Cheng Kung University Hospital, College of Medicine, National Cheng Kung University, Tainan 70101, Taiwan; ericababa@msn.com (J.-L.C.); chienchou.su@gmail.com (C.-C.S.); linwl@mail.ncku.edu.tw (W.-L.L.); 4School of Pharmacy & Institute of Clinical Pharmacy and Pharmaceutical Sciences, College of Medicine, National Cheng Kung University, Tainan 70101, Taiwan; 5Center of Infectious Disease and Signaling Research, National Cheng Kung University, Tainan 70101, Taiwan

**Keywords:** community-acquired pneumonia, pneumococcal pneumonia, bacterial pneumonia, 13-valent pneumococcal conjugate vaccine

## Abstract

The impact of the 13-valent pneumococcal conjugate vaccine (PCV13) on overall community-acquired pneumonia (CAP) and disease severity still needs thorough evaluation. In this study, we retrieve both pneumococcal CAP (P-CAP) and unspecific CAP (U-CAP) inpatient data from the Taiwan National Health Insurance Database (NHID) between 2005 and 2016. The interrupted time-series (ITS) analysis was performed to compare the incidence trend before and after the implementation of PCV13. After PCV13 implementation, there is a significant decreasing trend of P-CAP hospitalization, especially in children <1 year, 2–5 years, adults aged 19–65 years, 66 years, or older (all *p* value < 0.05). This corresponds to a 59% reduction in children <1 year, 47% in children aged 2–5 years, 39% in adult aged 19–65 years, and 41% in elderly aged 66 years or older. The intensive care rate (6.8% to 3.9%), severe pneumonia cases (21.7 to 14.5 episodes per 100,000 children–years), and the need for invasive procedures (4.3% to 2.0%) decreased in children aged 2–5 years (*p* value < 0.0001) with P-CAP. This PCV13 implementation program in Taiwan not only reduced the incidence of P-CAP, but also attenuated disease severity, especially in children aged 2–5 years.

## 1. Introduction

Pneumonia is the single largest infectious cause of death among all age populations worldwide. Approximately 150 million new pneumonia cases occur annually among children under 5 years of age worldwide, resulting in approximately 10–20 million hospitalizations [1]. Deaths due to pneumonia were recorded in 808,694 children in 2017, accounting for 15% of all deaths in children under five years old [2]. In the pre-vaccine era, *Streptococcus pneumoniae* was the leading pathogen in community-acquired pneumonia (CAP) and is likely associated with pulmonary complications, including parapneumonic effusion, empyema, lung abscess, pneumatocele, and necrotizing pneumonia. The 7-valent pneumococcal conjugate vaccine (PCV7), introduced in 2002, has dramatically decreased invasive pneumococcal disease (IPD), which most commonly manifests with occult bacteremia, bacteremic pneumonia, and meningitis [3]. The 13-valent pneumococcal conjugate vaccine (PCV13) later replaced PCV7 with six additional serotypes and especially includes serotype 19A, which is notorious for its antimicrobial resistance and pulmonary invasion in the post-PCV7 era. The PCV13 effectively eliminated the residual IPD caused by non-PCV7 serotypes, mainly 19A thereafter [4]. These conjugate pneumococcal vaccines not only decrease invasive disease, but also decrease nasopharyngeal pneumococcal carriage in vaccinated children, thereby causing less mucosal infection such as sinusitis, or otitis media [5]. In contrast to the evidence supporting the impact of the conjugate pneumococcal vaccine on IPD, its influence on non-invasive pneumococcal disease, especially overall CAP is relatively scarce, not to mention the possible influence on pneumonia severity or associated complications. 

In Taiwan, a national catch-up immunization program was launched in 2013, providing one dose of PCV13 for children aged 2–5 years, followed by two doses of PCV13 for children older than 1 year since 2014, and two-plus-one doses of PCV13 for children older than 2 months since 2015. The IPD incidence decreased steadily after the national vaccination program rolled out, leading to decreased medical expenses [6]. However, the impact of PCV13 on the incidence of pneumonia hospitalization and related clinical outcomes remains unexplored using this stepwise vaccination strategy. Therefore, we used the National Health Insurance Database (NHID) to evaluate the hospitalization rate and related clinical outcomes of pneumococcal CAP (P-CAP) before and after the stepwise implementation of PCV13 on a nationwide scale. Unspecific CAP (U-CAP) was used as an external comparison group to evaluate the specular trend of CAP cases in health care quality over time.

## 2. Materials and Methods

### 2.1. Data Source

This study was conducted using the National Health Insurance Database (NHID) in Taiwan. The National Health Insurance program is a single-payer and mandatory-enrolled system, and over 99% of Taiwan’s population (23 million residents) were covered in this system. Briefly, the NHID contains medical records, including ambulatory care, inpatients, prescriptions dispensed at pharmacies, and demographic data. Patient-level demographic information can be obtained through database linkage by anonymous, de-identified ID, or specific linkage codes. Detailed clinical information and cause of death from registry database can also be obtained. This research protocol was approved by the Institutional Review Board (IRB) of the National Cheng Kung University Hospital (NCKUH-B-ER-106-310).

### 2.2. Definitions of CAP Admissions and Clinical Outcomes

We retrieved the data of patients aged between 2 months and 100 years who were admitted to the hospital due to P-CAP (ICD-9: 481.XX; ICD-10: J13) or U-CAP (ICD-9 code: 486.XX, ICD-10: J18.9) with the corresponding ICD-9/10 codes from 2005 to 2016. In the clinical setting, physicians categorize pneumonia patients into pneumococcal CAP if there is microbiological evidence of pneumococcal infection. The unspecific CAP codes are applied if no causative pathogens are identified and usually account for most pneumonia cases. The validity of pneumonia diagnosis codes in the NHIRD was checked in the previous study, showing a sensitivity of 94.7% for inpatients and 92.3% for outpatients [7]. In addition, the data quality checked by the Bureau of NHI performs auditing reviews on a random sample one per 20 inpatient claims quarterly, and false reporting of diagnostic information results in a severe penalty from the Bureau [8]. Patients who had missing values for age or sex, single hospital stay of more than 90 days, or had both diagnoses (P-CAP and U-CAP) on the same admission, were excluded. Another admission episode was counted in patients with multiple admission episodes but the interval between each admission was longer than 90 days. Episodes with admission date in January 2005 and December 2016 were excluded if there were incomplete data presenting in winter 2004 and 2016. Patients were stratified into six age groups: 2 months to 1 year, 1 year, 2–5 years, 6–18 years, 19–65 years, and more than 65 years for comparison analysis. Clinical allocation and outcomes were retrieved from the database with corresponding codes and definitions listed below. Severe pneumonia was defined as any of the following conditions during admission: need for respiratory support (including intubation, ventilator use, non-invasive positive pressure support, and oxygen therapy), or pneumonia complications (including pleural effusion, empyema, necrotizing pneumonia, pneumatocele, lung abscess, respiratory failure, bacteremia/or sepsis, and hemolytic uremic syndrome). Complications requiring invasive procedures were retrieved using the reimbursement codes for thoracocentesis, exploratory thoracotomy, chest tube insertion, thoracoscopic decortication of the pleura, computed tomography-guided aspiration, or echo-guided aspiration. The details of the corresponding codes are presented in Appendix A.

### 2.3. Statistical Analyses

The hospitalization rate was calculated as the episodes of admissions per 100,000 inhabitants for each age group. For the interrupted time series analysis (ITS), we defined the pre-vaccination period from spring 2005 to winter 2012 and the post-vaccination period from spring 2013 to fall 2016 to analyze the trend of hospitalization rates. We estimated the hospitalization rate difference after the intervention in each age group based on the ITS model and seasonal effect was adjusted. Univariate analysis was used to evaluate the clinical outcomes. Student’s t-test was used for continuous variables, and the chi-square (χ^2^) test was used for categorical variables. All test hypotheses were evaluated using two-tailed t-tests. A *p*-value of <0.05 was considered statistical significance. Data extraction and all statistical analyses were performed using SAS software (version 9.4; SAS Institute Inc., Cary, NC, USA).

## 3. Results

### 3.1. The Trend of Hospitalization Rates before and after PVC13 Introduction

A total of 1,326,270 subjects that contributed 1,673,339 admissions were included in this study, of which 47,634 admissions (2.8%) were diagnosed with P-CAP and 1,625,705 (97.2%) were diagnosed with U-CAP (Figure 1). The total U-CAP hospitalization episodes were 34 times greater than P-CAP. The P-CAP and U-CAP yearly hospitalization rate among all ages is shown in Appendix A and Figure 2. For P-CAP, the hospitalization rate was highest among children aged 1 year (101.9–254.0 admissions per 100,000 person–years) and 2–5 years (109.3–232.9 admissions per 100,000 person–years), followed by children aged under 1 year (79.5–144.8 admission per 100,000 person–years) during the study period. For U-CAP, the hospitalization rate was highest among adults aged older than 66 years old (3020.4–3901.8 admissions per 100,000 person–years), followed by children aged 1 year (992.9–1734.2 admissions per 100,000 person–years). The ITS analysis revealed there was an increasing trend of P-CAP hospitalization in children, and U-CAP hospitalization in all age groups except children less than 2 years before intervention (Figure 3). After intervention, a downward trend of hospitalization was observed in P-CAP group, especially in children age under 5 years old. The estimated reduction in hospitalization rate after the vaccination program was mostly observed in the P-CAP group, but not in the U-CAP group (Table 1). The reduction in the P-CAP hospitalization rate was especially significant in children <1 year, 2–5 years, adult aged 19–65 years, 66 years, or older (all *p* value < 0.05). This corresponded to a 59% reduction in children <1 year, 47% in children aged 2–5 years, 39% in adult age 19–65 years, and 41% in elderly aged 66 years, or older. A similar trend was also noted in children aged 1 year but demonstrated only marginally significant (*p* value = 0.06). Although there was a seasonal distribution of CAP cases, the overall declining trend of P-CAP after vaccine implementation remained unchanged after adjusting the seasonal effects. The seasonality adjustment showed similar results (Appendix A). For U-CAP cases, there was no obvious decreasing trend of hospitalization after intervention in all age groups, except those aged 66 year or older. 

### 3.2. PCV13-Associated Reductions in the Clinical Outcomes

Four clinical outcomes with regard to P-CAP and U-CAP were summarized in Table 2. The length of hospital stay declined after the implementation of PCV13 in all age group with either P-CAP or U-CAP (all *p* value < 0.05), except the adult population with P-CAP. The ICU admission percentage dropped significantly in children aged under five with P-CAP (<1 year, 1 year, and 2–5 years age group, all *p* value < 0.05), but not with U-CAP. The percentage of severe pneumonia decreased from 19.6–1.7% to 14.5−16.3% in children aged 2–5 years and 6–18 years with P-CAP (*p* value < 0.001), but increased in the same age group with U-CAP (14.3–16.0% in the pre-vaccine period vs. 16.1−17.6% in the post-vaccine period, *p* value < 0.0001). Although the need for an invasive procedure was relatively low in the pediatric population with either P-CAP or U-CAP compared to the adult population, children with P-CAP required more invasive procedures (0.3–4.3%) than U-CAP (0.4–1.0%). Children aged 2–5 years with P-CAP had the most significant reduction in invasive procedures (2.3%, *p* value < 0.0001) among all CAP age groups after PCV13 intervention. Overall, the adult population with CAP had a longer hospital stay and higher chance of ICU admission, severe pneumonia, and the need for an invasive procedure compared to the pediatric population.

## 4. Discussion

In the current study, we used the NHID to establish the nationwide epidemiological data of P-CAP and U-CAP during 2005–2016 to evaluate the impact of the national immunization strategy. Our work demonstrated that before 2013, children under 5 years remained the most vulnerable population for P-CAP. However, after the implementation of the PCV13 vaccination program, there was a continuous decreasing trend in P-CAP cases in children using the interrupted time series analysis. This phenomenon was also accompanied by a decrease in hospital stay, severe pneumonia cases, the need for intensive care, and invasive procedures, validating the influence of PCV13 on decreasing P-CAP and attenuating disease severity and on a possibly improving health care quality as compared with U-CAP.

Given that pneumococcal disease is a vaccine-preventable disease which can be efficiently eradicated by vaccination, the World Health Organization (WHO) has recommended that pneumococcal vaccines (PCVs) should be included in all routine childhood immunization programs since 2006 [9]. Following this, the Advisory Committee for Immunization Practices (ACIP) of the United States Centers for Disease Control and Prevention (CDC) also recommended routine vaccination of all children aged 2–59 months with PCV13 in 2010 [10]. However, only a limited number of countries have included PCV13 in routine childhood immunization programs. Outside the routine pneumococcal infant immunization programs, the recommendation for PCV13 vaccination varies greatly among countries in terms of age groups and risk population [11]. In Taiwan, before PCV13 was adapted into the routine infant immunization program, the PCV coverage rate was low. In 2007, only 15.9% of children less than 5 years of age received one or more doses of PCV7 [6]. In order to eliminate IPD, the Taiwan Centers for Disease Control (Taiwan CDC) implemented a stepwise PCV13 vaccination program in 2013 based on the routine vaccination experience using the two-plus-one dose regimen, which was first implemented in Quebec, Canada [12]. This national catch-up primary vaccination program effectively increased the PCV coverage. The coverage rate for three doses of PCV13 for children born after January 2015 was approximately 95.8%, according to the Taiwan National Immunization Information System [13]. This strategy demonstrated great success in combating pneumococcal diseases in both children and adult populations in Taiwan. The IPD incidence in children aged 0–5 years decreased continuously from 18.9/100,000 in 2010–2012 to 9.4/100,000 in 2013–2014 and 6.3/100,000 in 2015–2017, corresponding to a 69% reduction [14]. 

In most countries, IPD usually attacks children under 2 years of age, with occult bacteremia as the most common clinical presentation. In Taiwan, children aged 2–5 years have a higher risk of IPD than those below 2 years of age and usually manifest with complicated pneumonia [14]. After the PCV13 vaccination program rolled out with extended coverage, the pediatric admission for pneumococcal empyema in children aged 2–5 years was 848.7 cases per 100,000 admissions in 2012, and then decreased to 238.4 cases per 100,000 admissions in 2013. In our previous multicenter community-acquired pneumonia study, we also found that the total number of pneumococcal pneumonia cases, especially those with definite microbiological evidence, decreased after 2013 [15]. However, changes in the overall incidence of pneumococcal pneumonia, not only that of complicated pneumonia, at the national scale after the introduction of PCV13 in Taiwan, are scarce. 

In countries with PCV universal vaccination, there is marginal evidence demonstrating a decrease in pneumococcal pneumonia cases. For example, in Sweden, a decrease in hospitalization for pneumonia in children under 5 years of age was found after universal PCV7 vaccination since 2007. Hospitalizations for pneumonia decreased significantly in children aged 0 to <2 years, from 450 to 366 per 100,000 population (relative risk (RR) = 0.81, *p* < 0.001) and in those aged 2 to <5 years from 250 to 212 per 100,000 population (RR = 0.85, *p* = 0.002). The sequential introduction of PCV7 and PCV13 led to a 19% lower risk of hospitalization for pneumonia in children aged 0–2 years [16]. In Scotland, routine PCV7 and PCV13 immunizations in young children were introduced in 2006 and 2010, respectively. They found that all-cause pneumonia hospitalization rates in children <2 years decreased by approximately 30% in the post-PCV-13 period compared with the pre-PCV period [17]. One meta-analysis study showed a reduction of 17% (95% CI: 11–22%, *p* < 0.001) for clinically confirmed pneumonia and 31% (95% CI: 26–35%, *p* < 0.001) for radiologically confirmed pneumonia in the hospitalization rates after the introduction of PCV10 or PCV13. In children aged 24–59 months, a reduction of 9% (95% CI: 5–14%, *p* < 0.001) and 24% (95% CI: 12–33%, *p* < 0.001) in the hospitalization rates for clinically and radiologically confirmed pneumonia were also demonstrated, respectively [18]. In our study, we demonstrated a decreasing trend of P-CAP in children aged 2–5 years after PCV13 implementation, but this was not significant in other age groups or U-CAP. However, it is not feasible to compare our data directly with other previous studies because of differences in pneumonia definitions and age stratifications. 

Similar to a previous study, we also found that children under five years had the highest P-CAP hospitalization rate among all age groups. The P-CAP hospitalization rate among children under five peaked during 2010–2012 (range: 509.7–631.7 admissions/100,000 children–years), then gradually declined since 2013. This trend parallels the incidence of national invasive pneumococcal disease (IPD), which also had a peak incidence during 2010–2012 [14]. During this period, pneumococcal serotype 19A surged to be the most important type of IPD, and the proportion of serotype 19A in children under five rose to 52.0–60.8%. This explains the increasing P-CAP hospitalization rate, given that serotype 19A infection usually manifests as complication P-CAP. Meanwhile, our data showed that the proportion of severe pneumonia and the need for invasive procedures, including chest tube insertion, thoracocentesis, and/or VATS in children aged 2–5 years decreased from 21.7% to 14.5% and 4.3% to 2.0%, respectively. This was also accompanied by a decrease in hospital stay and the need for intensive care, which again proved that PCV13 implementation targeted the major children age group accounting for most of the pneumococcal pneumonia cases and associated complications. Although we do not have the pneumococcal serotypes results in the NHIRD, the IPD national notification system demonstrated an obvious serotype replacement after the vaccine program roll-out [14]. Therefore, these pneumococcal CAP were likely to be caused by non-PCV13 serotypes and presented a milder clinical course. In our previous study, we also found that PCV-vaccinated patients who once obtained a breakthrough infection had a significantly higher lowest WBC, lower neutrophils, lower lymphocytes, and lower CRP values than non-vaccinated patients, indicating a less severe clinical disease [15]. In contrast, the clinical outcome was not present in the U-CAP group, especially in children aged 2–5 years. Overall, PCV13 vaccination not only decreased invasive or bacteremic pneumococcal CAP, but also decreased the overall pneumococcal CAP disease burden and attenuated disease severity.

In addition to the national PCV13 roll-out for children, the 23-valent pneumococcal polysaccharide vaccine (PPV23) was provided for the elder ≥75 years in Taiwan since 2008 and then extended to elderly 65–74 years in the following years. The PPV23 demonstrated a comparable vaccine efficacy of PCV13 against vaccine-serotype disease in the elderly. One meta-analysis showed PPV23 vaccine efficacy against IPD is 73% and 64% against pneumococcal pneumonia [19]. However, in the study conducted by Su, W.J. et al., the PPV23 uptake rate in Taiwan was only 7.2% in the elderly 65–74 years and 41.9% in the elderly >75 years during 2008–2016 [20]. In our study, the decline in pneumococcal CAP among adults ≥66 years from pre- to post-PCV13 immunization strategy was likely due to a combination of direct effects from PPV23 and indirect effects from PCV13, which were epidemiologically challenging to tease apart. As to the decline in the U-CAP among adults ≥66 years, one possible explanation is that the U-CAP cases were up-estimated due to a lack of a causative pathogen workup, and many cases without microbiological investigation were truly pneumococcal infections.

Our study had some strengths. First, our findings in the present study were the first to use the nationwide health insurance database to evaluate the total P-CAP disease burden, along with U-CAP cases after the innovative stepwise two-plus-one PCV13 vaccination strategy. The estimation of the population-based incidence of pneumococcal pneumonia has been challenging and is, frequently, underestimated because the *S. pneumoniae* culture yield rate from blood or pleural effusion was very low, especially with prior antimicrobial treatment. Non-invasive diagnostic methods, including a sputum culture or urinary pneumococcal antigen test, could lead to false-positive results due to a high pneumococcal nasopharyngeal carriage in children [21,22]. However, the crude incidence rate of overall bacterial pneumonia and pneumococcal pneumonia from population-based databases was worthy of exploration, given that this would provide an overall glance of the disease burden from a nationwide perspective. Second, the NHID provided a powerful and accurate evaluation because nearly all populations were reimbursed in this system. Missing data on hospitalization for pneumonia were small and did not violate the assumption of linearity of the interrupted time series in the pre-vaccination and post-vaccine periods. Third, we had a long-term follow-up period to evaluate the impact of the intervention and used U-CAP as a comparison population.

However, there were some limitations to our study. First, the study data were retrieved through ICD codes, which could not precisely reflect the causative pathogens if the cases did not undergo a complete microbiological investigation or the clinician failed to make an accurate diagnosis. Bacterial pneumonia may have been over-diagnosed for the intention to prescribe antibiotics in most clinical conditions. The process of talking about causative pathogens and microbiological research to find out the microbiological etiology of pneumonias is complicated, as, despite many improvements, in nearly half of the pneumonia cases the pathogen is not detected [23]. In fact, in Figure 2, there was a peak of admissions in 2010 due to P-CAP and U-CAP, suggesting that many of the U-CAP were possible pneumococcal CAP. It was estimated that around 20% of pneumonia requiring admission were due to *S. pneumoniae*, what was much lower than the data in Figure 2 [24]. However, in a USA Nationwide Inpatient Sample study, the pneumococcal and all-cause CAP hospitalization rate was 13.9 and 447.4 per 100,000 persons, respectively. The proportion of pneumococcal CAP in all-cause CAP was 3.1%, comparable to our study [25]. This low pneumococcal CAP proportion in overall CAP by a large database study revealed that, in reality, the pathogen detection among pneumonia patients is far below expectation. Second, we did not know the living crowdedness, which is an important risk factor for pneumonia. Third, we did not evaluate the difference in physician practice behavior along with the change in pneumonia epidemiology over time. There might be some physician prejudice after the roll-out of the PCV13 vaccination program, assuming the pneumococcus is no longer one of the priority considerations in pneumonia. This might also affect their practice behaviors, both in diagnosis and prescription preferences. Finally, improving sanitary conditions, childcare, and less child-to-child transmission due to decreasing children’s population may also contribute to the decreasing pneumonia incidence in Taiwan. However, we found that the severity of U-CAP did not change in children aged 2–5 years.

In conclusion, the stepwise implementation of the PCV13 vaccination program had effectively decreased the incidence of P-CAP hospitalization, especially in children aged 2–5 years. This was accompanied by a decrease in overall hospital stays, severe pneumonia cases, the need for intensive care, and invasive procedures. This unique vaccination approach not only controls IPD or IPD-related complication pneumonia, but also extends its protection to overall P-CAP.

## Figures and Tables

**Figure 1 vaccines-09-01043-f001:**
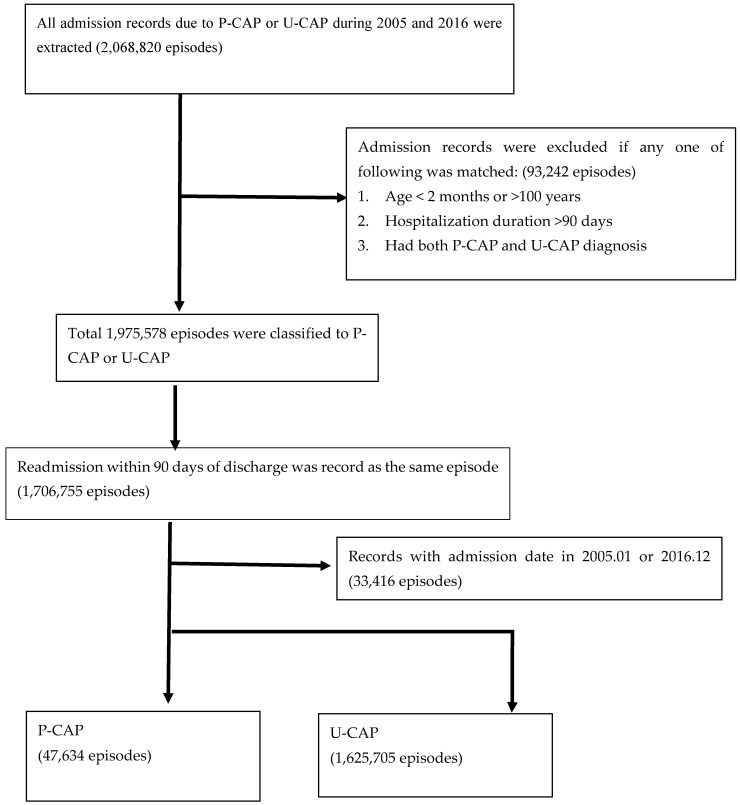
Analytic flowchart of pneumococcal and unspecific community-acquired pneumonia hospitalization from National Health Insurance Database during 2005 and 2016.

**Figure 2 vaccines-09-01043-f002:**
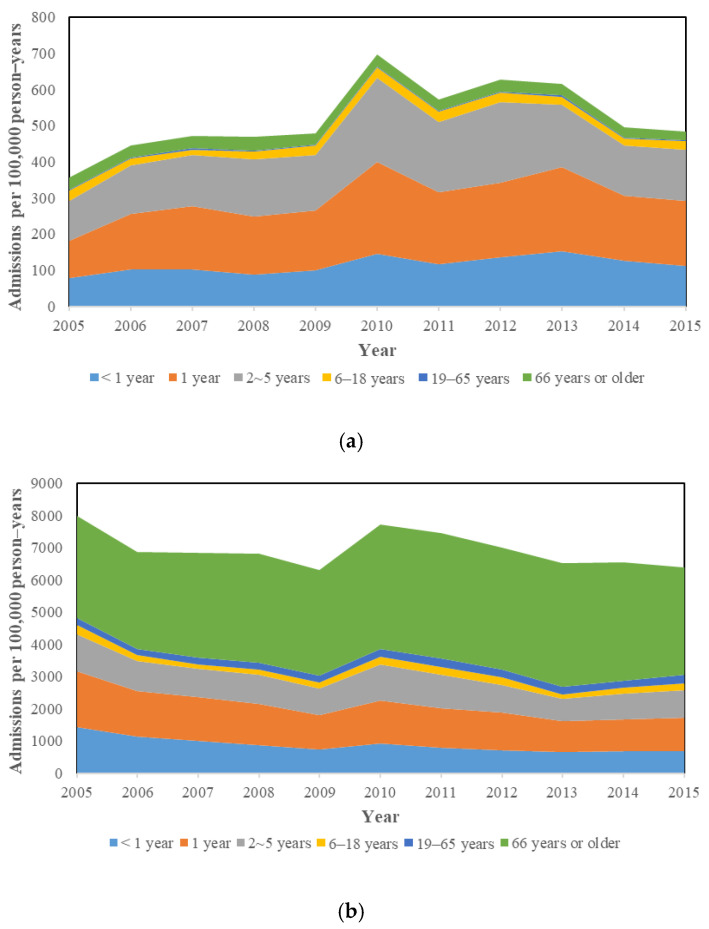
The hospitalization incidence rate of (**a**) pneumococcal and (**b**) unspecific community-acquired pneumonia during 2005–2015 by age group.

**Figure 3 vaccines-09-01043-f003:**
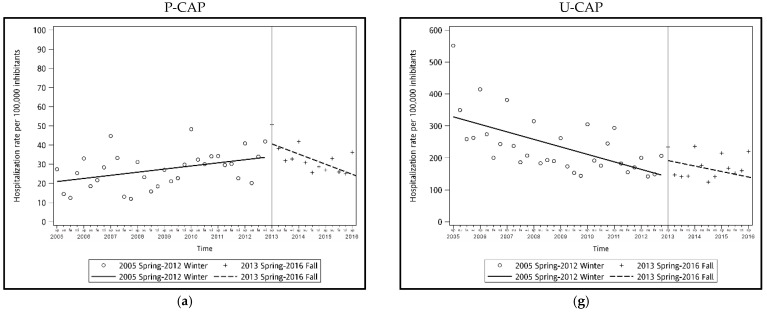
Trends of hospitalization rate due to pneumococcal and unspecific community-acquired pneumonia before and after vaccination. Trends in hospitalization rates before (hollow dots and solid trend line) and after (plus sign and broken trend line) vaccination program roll-out in 2013. For pneumococcal CAP (P-CAP) in children aged (**a**) <1 year, (**b**) 1 year, and (**c**) 2–5 years, (**d**) 6–18 years, adults aged (**e**) 19–65 years, (**f**) 66 years, or older, and unspecific CAP (U-CAP) in children aged (**g**) <1 year, (**h**) 1 year, and (**i**) 2–5 years, (**j**) 6–18 years, adults aged (**k**) 19–65 years, (**l**) 66 years, or older. The vertical line indicates the PCV13 intervention in 2013. The labels of x-axis: sp—spring; su—summer; fa—fall; wi—winter.

**Table 1 vaccines-09-01043-t001:** Estimated difference of hospitalization incidence rate between pre- (2005–2012) and post-vaccination period (2013–2016) among patients with pneumococcal or unspecific community-acquired pneumonia by age groups.

	P-CAP				U-CAP			
Age Groups	Estimated Difference of Hospitalization Rate ^1^	95% CI	*p* Value	% of Change	Estimated Difference of Hospitalization Rate	95% CI	*p* Value	% of Change
<1 year	−22.51	−35.52 to −9.51	<0.01	59%	20.50	−91.49 to 135.64	0.71	12%
1 year	−20.81	−41.55 to −0.08	0.06	39%	62.13	−85.94 to 219.76	0.4	27%
2~5 years	−18.01	−31.63 to −4.4	0.01	47%	39.84	−61.84 to 147.66	0.43	24%
6–18 years	−1.97	−5.21 to 1.26	0.24	35%	22.31	−9.42 to 57.46	0.17	64%
19–65 years	−0.31	−0.58 to −0.03	0.04	39%	−2.93	−19.38 to 13.08	0.71	5%
66 years or older	−3.25	−5.11 to −1.4	0 < 0.01	41%	−176.27	−367.43 to −12.22	0.04	20%

P-CAP: pneumococcal community-acquired pneumonia; U-CAP: unspecific community-acquired pneumonia. ^1^ hospitalization rate: admission episodes per 100,000 person–years.

**Table 2 vaccines-09-01043-t002:** Difference of clinical allocation and outcome between pre- (2005–2012) and post-vaccination period (2013–2016) among patients with pneumococcal and unspecific community-acquired pneumonia by age groups.

Age Groups	Length of Hospital Stay (Days, Mean ± SD)			ICU Admission (%)			Severe Pneumonia (%)			Invasive Procedure (%)		
P-CAP Group	Pre-	Post-	*p*-Value	Pre-	Post-	*p*-Value	Pre-	Post-	*p*-Value	Pre-	Post-	*p*-Value
**<1 year**	6.5 (5.6)	5.5 (3.8)	<0.0001	8.6	6.0	0.0187	25.2	24.1	0.5248	0.3	0.0	0.0921
**1 year**	5. 9 (4.3)	5.1 (3.2)	<0.0001	5.7	3.9	0.0067	19.2	17.4	0.1488	2.4	0.6	<0.0001
**2~5 years**	6.0 (4.4)	5.1 (3.3)	<0.0001	6. 8	3.9	<0.0001	21.7	14.5	<0.0001	4.3	2.0	<0.0001
**6** **–** **18 years**	5.9 (4.7)	5.4 (4.3)	<0.0001	6.3	5.1	0.2333	19.6	16.3	0.0002	2.9	1.9	0.0098
**19** **–** **65 years**	11.8 (12.1)	12.0 (12.4)	0.4708	22.5	24.4	0.1143	59.4	63.6	0.0027	9.4	9.8	0.6381
**66 years or older**	13. 9 (12. 9)	13.7 (12.2)	0.4143	29.9	27.4	0.0231	81.9	81.6	0.7571	11.1	9.7	0.0608
**U-CAP group**												
**<1 year**	7.1 (7.5)	6.3 (6.3)	<0.0001	11.2	10.3	0.0806	27.5	26.1	0.0545	0.5	0.5	0.969
**1 year**	5.5 (4.7)	5.1 (4.3)	<0.0001	5.0	4.6	0.2084	16.3	18.0	0.0005	0.4	0.3	0.3849
**2~5 years**	5.0 (3.7)	4.7 (3.7)	<0.0001	2.8	3.0	0.1067	14.3	16.1	<0.0001	0.6	0.3	<0.0001
**6** **–** **18 years**	5.5 (5.9)	5.3 (5.4)	<0.0001	4.4	4.3	0.2516	16.0	17.6	<0.0001	1.0	0.7	0.0026
**19** **–** **65 years**	14.2 (15.8)	13.1 (14.5)	<0.0001	24.9	21.7	<0.0001	61.1	60.7	0.0109	10.1	8.9	<0.0001
**66 years or older**	17.2 (16.8)	14.9 (14.4)	<0.0001	34.3	25.9	<0.0001	82.3	80.4	<0.0001	11.0	9.1	<0.0001

P-CAP: pneumococcal community-acquired pneumonia; U-CAP: unspecific community-acquired pneumonia. SD: standard deviation; ICU: intensive care unit.

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
