# Peer review of "Decrease of Pneumococcal Community-Acquired Pneumonia Hospitalization and Associated Complications in Children after the Implementation of the 13-Valent Pneumococcal Conjugate Vaccine (PCV13) in Taiwan"

_vaccines, 2021, doi:10.3390/vaccines9091043_

Round 1

Reviewer 1 Report

Authors have studied the impact of introduction of PCV13 in Taiwan on the incidence and clinical severity of CAP. Although the title says "step-wide" implementation, I assume "step-wise" implementation is meant.

The strength of the study is that a nationwide database is used, but one of the limitations is that pneumococcal CAP as well as unspecified CAP is extremely poorly defined. This is reflected in the proportion of pneumococcal CAP: less than 3% of all CAP cases. This is approx 10-fold lower than reported in literature. Authors should address this discrepancy.

Second major issue is the period of analysis (2005-2016). Full implementation of PCV13 (2+1 in 2 months old babies) was not started before 2015. This means that the < 1 years old group was analyzed for at best 1 year. I would strongly suggest to also include at least 2017 and 2018 in the analysis.

A remarkable finding is the reduction in length of hospital stay for pneumococcal CAP after implementation of PCV13. Where these cases with non-PCV13 CAP? and if so, what would cause the less severe CAP. If it would be cases of vaccinated children with PCV13 serotypes, they should be considered vaccine failures, as discussed in that context (breakthrough infection with less severe clinical course??).

In Figure 3 the X-axis is not labeled. The scattered data could indicate some seasonal effect in incidence; is that real? If so, please comment.

Author Response

Authors have studied the impact of introduction of PCV13 in Taiwan on the incidence and clinical severity of CAP. Although the title says "step-wide" implementation, I assume "step-wise" implementation is meant.

Reply to comment (1)

We had removed the step-wide from title to avoid confusion and revised to “stepwise” implementation in the manuscript. 

Comment (2)

The strength of the study is that a nationwide database is used, but one of the limitations is that pneumococcal CAP as well as unspecified CAP is extremely poorly defined. This is reflected in the proportion of pneumococcal CAP: less than 3% of all CAP cases. This is approx 10-fold lower than reported in literature. Authors should address this discrepancy.

Reply to comment (2) add a paragraph into the method and discussion

We had re-addressed the pneumococcal and unspecific CAP in detail in the method accordingly. Please refer to line 83-91. We also have added statement in the difference of the proportion of pneumococcal pneumonia in all pneumonia among different studies in the discussion. Please refer to line 324-329.

“In the clinical setting, physicians categorize pneumonia patients into pneumococcal CAP if there is microbiological evidence of pneumococcal infection. The unspecific CAP codes are applied if no causative pathogens are identified and usually account for most pneumonia cases. The validity of pneumonia diagnosis codes in the NHIRD was checked in the previous study, showing a sensitivity of 94.7% for inpatients and 92.3% for outpatients [7]. In addition, the data quality checked by the Bureau of NHI performs auditing reviews on a random sample one per 20 inpatient claims quarterly, and false reporting of diagnostic information results in a severe penalty from the Bureau [8].

“However, in a USA Nationwide Inpatient Sample study, the pneumococcal and all-cause CAP hospitalization rate is 13.9 and 447.4 per 100,000 persons, respectively. The proportion of pneumococcal CAP in all-cause CAP is 3.1%, comparable to our study [25]. This low pneumococcal CAP proportion in overall CAP by large database study revealed that, in reality, the pathogen detection among pneumonia patients is far below expectation.”

Comment (3)

Second major issue is the period of analysis (2005-2016). Full implementation of PCV13 (2+1 in 2 months old babies) was not started before 2015. This means that the < 1 years old group was analyzed for at best 1 year. I would strongly suggest to also include at least 2017 and 2018 in the analysis.

Reply to comment (3) 

We agree with the reviewer’s concern. The longer surveillance and monitor after the implementation of vaccine strategy is extremely important to evaluate the impact of PCV13, especially in children <1 year. However, there will be at least a lag time of 2 years in the availability of NHID [1] and user can no longer extend access once the project is closed. Although the follow up period for children < 1 year is a bit short, there is significant reduction of P-CAP using ITS analysis. Second, in another national study in Taiwan, the invasive pneumococcal disease (IPD) in children < 1 year decreased from 12-17 episodes per 100, 000 persons during 2008-2013 to less than 5 episodes per 100, 000 persons in 2015-2016, and continue to decline after 2017 [2]. This trend goes parallel with our findings in pneumococcal CAP and re-validates the impact of PCV13 on both IPD and overall P-CAP.    

  1. Hsieh CY, Su CC, Shao SC, Sung SF, Lin SJ, Kao Yang YH, Lai EC. Taiwan's National Health Insurance Research Database: past and future. Clin Epidemiol. 2019 May 3;11:349-358. doi: 10.2147/CLEP.S196293. PMID: 31118821.
  2. Lu CY, Chiang CS, Chiu CH, Wang ET, Chen YY, Yao SM, Chang LY, Huang LM, Lin TY, Chou JH. Successful Control of Streptococcus pneumoniae 19A Replacement With a Catch-up Primary Vaccination Program in Taiwan. Clin Infect Dis. 2019 Oct 15;69(9):1581-1587. doi: 10.1093/cid/ciy1127. PMID: 30923808.

Comment (4)

A remarkable finding is the reduction in length of hospital stay for pneumococcal CAP after implementation of PCV13. Where these cases with non-PCV13 CAP? and if so, what would cause the less severe CAP. If it would be cases of vaccinated children with PCV13 serotypes, they should be considered vaccine failures, as discussed in that context (breakthrough infection with less severe clinical course??).

Reply to comment (4) and add a paragraph in the discussion

We had added a paragraph in the discussion and amended the manuscript accordingly. Please refer to line 271-278

“Although we don’t have the pneumococcal serotypes result in the NHIRD, the IPD national notification system demonstrated an obvious serotype replacement after vaccine program roll out [12]. Therefore, these pneumococcal CAP were likely to be caused by non-PCV13 serotypes and presented a milder clinical course. In our previous study, we also found that PCV vaccinated patients who once got breakthrough infection had significant higher lowest WBC, lower neutrophils, lower lymphocytes, and lower CRP values than non-vaccinated patients, indicating a less severe clinical disease [15].” 

Comment (5)

In Figure 3 the X-axis is not labeled. The scattered data could indicate some seasonal effect in incidence; is that real? If so, please comment.

Reply to comment (5)

The incidence rate was higher in spring and winter. The seasonal effect was adjusted, and the related method and result were presented in method part and supplementary table3, respectively. In addition, we labeled the X-axis in Figure 3.  

Reviewer 2 Report

Article describing the clinical influence of introduction of PCV13 in P-CAP and U-CAP in Taiwan. The article adds many valuable information on other relevant (and less described) aspects of the beneficial effects of PCV13, as days of admission and severity of disease, although unfortunately no data on the effect on mortality is given. The article is well written, is easy to follow and results support the main conclusions drawn by authors. Besides, as it is Nationwide study, the number of cases included is very big what can mitigate the errors in the assignation of ICD codes.

Major comments

In the first limitation of the study, the authors could also add, when taking about causative pathogens and microbiological research (lines 294-2969) that to find out the microbiological etiology of pneumonias is complicated, as despite many improvements, in nearly half of the pneumonia cases the pathogen is not detected (1). In fact, in figure 2, there is a peak of Admissions in 2010due to P-CAP and U-CAP, suggesting that many of the U-CAP are indeed pneumococcal pneumonia. It is estimated that around 20% of pneumonia requiring admission are due to S. pneumoniae (2), what is much lower than the data in Figure 2.

  1. Mandell LA, Wunderink RG, Anzueto A, et al. Infectious Diseases Society of America/American Thoracic Society consensus guidelines on the management of community-acquired pneumonia in adults. Clin Infect Dis. 2007;44 Suppl 2:S27-72. doi: 10.1086/511159.
  2. Cilloniz C, Martin-Loeches I, Garcia-Vidal C, San Jose A, Torres A. Microbial Etiology of Pneumonia: Epidemiology, Diagnosis and Resistance Patterns. Int J Mol Sci. 2016;17:2120. doi: 10.3390/ijms17122120.

Discussion. In my opinion, part of the discussion is based more on data from other works than on the data provided in the results section. I agree that it is important to compare and provide data and information from other studies carried out in Taiwan. However, the text between lines 191 to 236 is an interesting review on IPD and distribution/replacement of serotypes with serotypes 3 and 19A in Taiwan, but there is no reference or comparison with the results obtained in the present study, which nevertheless does not show at all that the changes observed were due to certain serotypes.

Besides, there is no analysis on the U-CAP data in the discussion, nor on the group protection given by PCV13 to those over 66 years of age, which could be very interesting for readers, given the importance and burden of pneumonia in the elderly. In fact, the significant reductions in the hospitalizations rate of P-CAP and especially U-CAP aster vaccination in >66-years is quite striking and deserve an analysis.

Minor comments

Authors. There are two commas after “Wen-Liang Lin3, ,”

Material and Methods. The study is quite long (12 years) but it would be desirable to know, approximately, the data of population in each of the 5 age-groups of the study (for instance in 2010) to compare with other countries.

Figure 1. Could the author point out the number of cases corresponding to each of the 93,242 excluded episodes of <2 months, admission >90 days and had both CAP diagnosis?

Lines 282-285. “Non-invasive diagnostic methods, including sputum culture or urinary pneumococcal antigen test, could lead to false positive results due to high pneumococcal nasopharyngeal carriage in children.” Please, give a reference for this statement.

Line 309-311. “In conclusion, the step-wide implementation of the PCV13 vaccination program in 2013 demonstrated great success in combating IPD and eliminating the locally prevalent pneumococcal serotype (19A) in a previous study.”

Although the sentence is correct, this is not a conclusion of this study that has not determined IPD neither serotypes causing pneumococcal disease.

Author Response

Article describing the clinical influence of introduction of PCV13 in P-CAP and U-CAP in Taiwan. The article adds many valuable information on other relevant (and less described) aspects of the beneficial effects of PCV13, as days of admission and severity of disease, although unfortunately no data on the effect on mortality is given. The article is well written, is easy to follow and results support the main conclusions drawn by authors. Besides, as it is Nationwide study, the number of cases included is very big what can mitigate the errors in the assignation of ICD codes.

Major comment (1)

In the first limitation of the study, the authors could also add, when taking about causative pathogens and microbiological research (lines 294-2969) that to find out the microbiological etiology of pneumonias is complicated, as despite many improvements, in nearly half of the pneumonia cases the pathogen is not detected (1). In fact, in figure 2, there is a peak of Admissions in 2010due to P-CAP and U-CAP, suggesting that many of the U-CAP are indeed pneumococcal pneumonia. It is estimated that around 20% of pneumonia requiring admission are due to S. pneumoniae (2), what is much lower than the data in Figure 2.

(1) Mandell LA, Wunderink RG, Anzueto A, et al. Infectious Diseases Society of America/American Thoracic Society consensus guidelines on the management of community-acquired pneumonia in adults. Clin Infect Dis. 2007;44 Suppl 2:S27-72. doi: 10.1086/511159.

(2) Cilloniz C, Martin-Loeches I, Garcia-Vidal C, San Jose A, Torres A. Microbial Etiology of Pneumonia: Epidemiology, Diagnosis and Resistance Patterns. Int J Mol Sci. 2016;17:2120. doi: 10.3390/ijms17122120.

Response to comment (1) and add a paragraph in the discussion

We had added the limitation of this study according to your suggestions and amended the manuscript accordingly. Please refer to line 318-324.

Major comment (2)

Discussion. In my opinion, part of the discussion is based more on data from other works than on the data provided in the results section. I agree that it is important to compare and provide data and information from other studies carried out in Taiwan. However, the text between lines 191 to 236 is an interesting review on IPD and distribution/replacement of serotypes with serotypes 3 and 19A in Taiwan, but there is no reference or comparison with the results obtained in the present study, which nevertheless does not show at all that the changes observed were due to certain serotypes.

Response to comment (2)

We had deleted part of the discussion addressing the serotype replacement (previous line 213-215, 229-239) and amended the manuscript accordingly.

Major Comment (3)

Besides, there is no analysis on the U-CAP data in the discussion, nor on the group protection given by PCV13 to those over 66 years of age, which could be very interesting for readers, given the importance and burden of pneumonia in the elderly. In fact, the significant reductions in the hospitalizations rate of P-CAP and especially U-CAP aster vaccination in >66-years is quite striking and deserve an analysis.

Response to comment (3) and add a paragraph in the discussion

We had added the discussion of protective effect on elderly and the reduction of hospitalization of P-CAP accordingly. Please refer to line 283-296.

“In addition to the national PCV13 roll-out for children, the 23-valent pneumococcal polysaccharide vaccine (PPV23) was provided for the elder ³ 75 years in Taiwan since 2008 and then extended to elderly 65-74 years in the following years. The PPV23 demonstrated a comparable vaccine efficacy of PCV13 against vaccine-serotype disease in the elderly. One meta-analysis showed PPV23 vaccine efficacy against IPD is 73% and 64% against pneumococcal pneumonia [19]. However, in the study conducted by WJ Su et al., the PPV23 uptake rate in Taiwan is only 7.2% in the elderly 65-74 years and 41.9% in the elderly >75 years during 2008-2016 [20]. In our study, the decline in pneumococcal CAP among adults ≥ 66 years from pre- to post-PCV13 immunization strategy is likely due to a combination of direct effects from PPV23 and indirect effects from PCV13, which are epidemiologically challenging to tease apart. As to the decline in the U-CAP among adults ≥ 66 years, one possible explanation is that the U-CAP cases are up-estimated due to lack of causative pathogen workup, and many cases without microbiological investigation are truly pneumococcal infections.”

Minor comments

Authors. There are two commas after “Wen-Liang Lin3, ,”

Response to this comment:

We had removed the comma.

Minor comments (1)

Material and Methods. The study is quite long (12 years) but it would be desirable to know, approximately, the data of population in each of the 5 age-groups of the study (for instance in 2010) to compare with other countries.

Response to comment (1)

For Taiwan has become an aging society in 1993, the proportion of children was slightly decreased yearly. Comparing the population in 2013, the year of PCV-13 vaccine program introduction, with the WHO Standard Population, the proportion of children less than 14 yrs was lower and the elderly was higher in Taiwan population (As table showed below, SMD> ±0.1 means significant difference).

Age Group

WHO World Standard (%)

Taiwan (%)

SMD

0-4

8.86

4.22

-0.188476

5-9

8.69

4.52

-0.168701

10-14

8.6

5.74

-0.11108

15-19

8.47

6.79

-0.063121

20-24

8.22

6.89

-0.050323

25-29

7.93

7.15

-0.029492

30-34

7.61

8.68

0.039168

35-39

7.15

8.14

0.037104

40-44

6.59

7.82

0.047724

45-49

6.04

7.99

0.076469

50-54

5.37

7.93

0.102841

55-59

4.55

7.09

0.108793

60-64

3.72

5.69

0.093287

65-69

2.96

11.34

0.019394

70-74

2.21

2.95

0.046951

75-79

1.52

2.21

0.051183

80-84

0.91

1.63

0.063953

85-89

0.44

0.88

0.054586

90-94

0.15

0.30

0.03098

95-99

0.04

0.06

0.009109

100+

0.005

0.01

0.007286

Total

100.035

100.00

Minor comments (2)

Figure 1. Could the author point out the number of cases corresponding to each of the 93,242 excluded episodes of <2 months, admission >90 days and had both CAP diagnosis?

Response to comment (2)

For these criteria were not mutually exclusive in this study, only the total number could be presented. 

Minor Comment (3)

Lines 282-285. “Non-invasive diagnostic methods, including sputum culture or urinary pneumococcal antigen test, could lead to false positive results due to high pneumococcal nasopharyngeal carriage in children.” Please, give a reference for this statement.

Response to comment (3)

The reference below is added into the manuscript accordingly.

21.García-Vázquez E, Marcos MA, Mensa J, et al. Assessment of the Usefulness of Sputum Culture for Diagnosis of Community-Acquired Pneumonia Using the PORT Predictive Scoring System. Arch Intern Med. 2004;164(16):1807–1811. doi:10.1001/archinte.164.16.1807

  1. Domínguez J, Blanco S, Rodrigo C, et al. Usefulness of urinary antigen detection by an immunochromatographic test for diagnosis of pneumococcal pneumonia in children. J Clin Microbiol. 2003;41(5):2161-2163. doi:10.1128/JCM.41.5.2161-2163.2003

Minor comment (4)

Line 309-311. “In conclusion, the step-wide implementation of the PCV13 vaccination program in 2013 demonstrated great success in combating IPD and eliminating the locally prevalent pneumococcal serotype (19A) in a previous study.”

Although the sentence is correct, this is not a conclusion of this study that has not determined IPD neither serotypes causing pneumococcal disease.

Response to comment (4)

We had deleted this sentence amended this paragraph accordingly. Please refer to line 339-340.

“ In conclusion, the stepwise implementation of the PCV13 vaccination program had effectively decreased the incidence of P-CAP hospitalization, especially in children aged 2–5 years……..”

Round 2

Reviewer 1 Report

My last comment, #5 has not been addressed satisfactorily. 

Comment (5)

In Figure 3 the X-axis is not labeled. The scattered data could indicate some seasonal effect in incidence; is that real? If so, please comment.

The revised X-axis of Figure3 now indicates the range, but that is not proper labeling. Please use major tick-marks for the year and minor tick marks per season.
